# DISTRIBUTED PRIORITIZED EXPERIENCE REPLAY

**Dan Horgan**
DeepMind
horgan@google.com

**John Quan**
DeepMind
johnquan@google.com

**David Budden**
DeepMind
budden@google.com

**Gabriel Barth-Maron**
DeepMind
gabrielbm@google.com

**Matteo Hessel**
DeepMind
mtthss@google.com

**Hado van Hasselt**
DeepMind
hado@google.com

**David Silver**
DeepMind
davidsilver@google.com

## ABSTRACT

We propose a distributed architecture for deep reinforcement learning at scale, that enables agents to learn effectively from orders of magnitude more data than previously possible. The algorithm decouples acting from learning: the actors interact with their own instances of the environment by selecting actions according to a shared neural network, and accumulate the resulting experience in a shared experience replay memory; the learner replays samples of experience and updates the neural network. The architecture relies on prioritized experience replay to focus only on the most significant data generated by the actors. Our architecture substantially improves the state of the art on the Arcade Learning Environment, achieving better final performance in a fraction of the wall-clock training time.

## 1 INTRODUCTION

A broad trend in deep learning is that combining more computation (Dean et al., 2012) with more powerful models (Kaiser et al., 2017) and larger datasets (Deng et al., 2009) yields more impressive results. It is reasonable to hope that a similar principle holds for deep reinforcement learning. There are a growing number of examples to justify this optimism: effective use of greater computational resources has been a critical factor in the success of such algorithms as Gorila (Nair et al., 2015), A3C (Mnih et al., 2016), GPU Advantage Actor Critic (Babaeizadeh et al., 2017), Distributed PPO (Heess et al., 2017) and AlphaGo (Silver et al., 2016).

Deep learning frameworks such as TensorFlow (Abadi et al., 2016) support distributed training, making large scale machine learning systems easier to implement and deploy. Despite this, much current research in deep reinforcement learning concerns itself with improving performance within the computational budget of a single machine, and the question of how to best harness more resources is comparatively underexplored.

In this paper we describe an approach to scaling up deep reinforcement learning by generating more data and selecting from it in a prioritized fashion (Schaul et al., 2016). Standard approaches to distributed training of neural networks focus on parallelizing the computation of gradients, to more rapidly optimize the parameters (Dean et al., 2012). In contrast, we distribute the generation and selection of experience data, and find that this alone suffices to improve results. This is complementary to distributing gradient computation, and the two approaches can be combined, but in this work we focus purely on data-generation.

We use this distributed architecture to scale up variants of Deep Q-Networks (DQN) and Deep Deterministic Policy Gradient (DDPG), and we evaluate these on the Arcade Learning Environment benchmark (Bellemare et al., 2013), and on a range of continuous control tasks. Our architecture

achieves a new state of the art performance on Atari games, using a fraction of the wall-clock time compared to the previous state of the art, and without per-game hyperparameter tuning.

We empirically investigate the scalability of our framework, analysing how prioritization affects performance as we increase the number of data-generating workers. Our experiments include an analysis of factors such as the replay capacity, the recency of the experience, and the use of different data-generating policies for different workers. Finally, we discuss implications for deep reinforcement learning agents that may apply beyond our distributed framework.

## 2 BACKGROUND

**Distributed Stochastic Gradient Descent**    Distributed stochastic gradient descent is widely used in supervised learning to speed up training of deep neural networks, by parallelizing the computation of the gradients used to update their parameters. The resulting parameter updates may be applied synchronously (Krizhevsky, 2014) or asynchronously (Dean et al., 2012). Both approaches have proven effective and are an increasingly standard part of the deep learning toolbox. Inspired by this, Nair et al. (2015) applied distributed asynchronous parameter updates and distributed data generation to deep reinforcement learning. Asynchronous parameter updates and parallel data generation have also been successfully used within a single-machine, in a multi-threaded rather than a distributed context (Mnih et al., 2016). GPU Asynchronous Actor-Critic (GA3C; Babaeizadeh et al., 2017) and Parallel Advantage Actor-Critic (PAAC; Clemente et al., 2017) adapt this approach to make efficient use of GPUs.

**Distributed Importance Sampling**    A complementary family of techniques for speeding up training is based on variance reduction by means of importance sampling (cf. Hastings, 1970). This has been shown to be useful in the context of neural networks (Hinton, 2007). Sampling non-uniformly from a dataset and weighting updates according to the sampling probability in order to counteract the bias thereby introduced can increase the speed of convergence by reducing the variance of the gradients. One way of doing this is to select samples with probability proportional to the $L_2$ norm of the corresponding gradients. In supervised learning, this approach has been successfully extended to the distributed setting (Alain et al., 2015). An alternative is to rank samples according to their latest known loss value and make the sampling probability a function of the rank rather than of the loss itself (Loshchilov & Hutter, 2015).

**Prioritized Experience Replay**    Experience replay (Lin, 1992) has long been used in reinforcement learning to improve data efficiency. It is particularly useful when training neural network function approximators with stochastic gradient descent algorithms, as in Neural Fitted Q-Iteration (Riedmiller, 2005) and Deep Q-Learning (Mnih et al., 2015). Experience replay may also help to prevent overfitting by allowing the agent to learn from data generated by previous versions of the policy. Prioritized experience replay (Schaul et al., 2016) extends classic prioritized sweeping ideas (Moore & Atkeson, 1993) to work with deep neural network function approximators. The approach is strongly related to the importance sampling techniques discussed in the previous section, but using a more general class of biased sampling procedures that focus learning on the most 'surprising' experiences. Biased sampling can be particularly helpful in reinforcement learning, since the reward signal may be sparse and the data distribution depends on the agent's policy. As a result, prioritized experience replay is used in many agents, such as Prioritized Dueling DQN (Wang et al., 2016), UNREAL (Jaderberg et al., 2017), DQfD (Hester et al., 2017), and Rainbow (Hessel et al., 2017). In an ablation study conducted to investigate the relative importance of several algorithmic ingredients (Hessel et al., 2017), prioritization was found to be the most important ingredient contributing to the agent's performance.

## 3 OUR CONTRIBUTION: DISTRIBUTED PRIORITIZED EXPERIENCE REPLAY

In this paper we extend prioritized experience replay to the distributed setting and show that this is a highly scalable approach to deep reinforcement learning. We introduce a few key modifications that enable this scalability, and we refer to our approach as *Ape-X*.

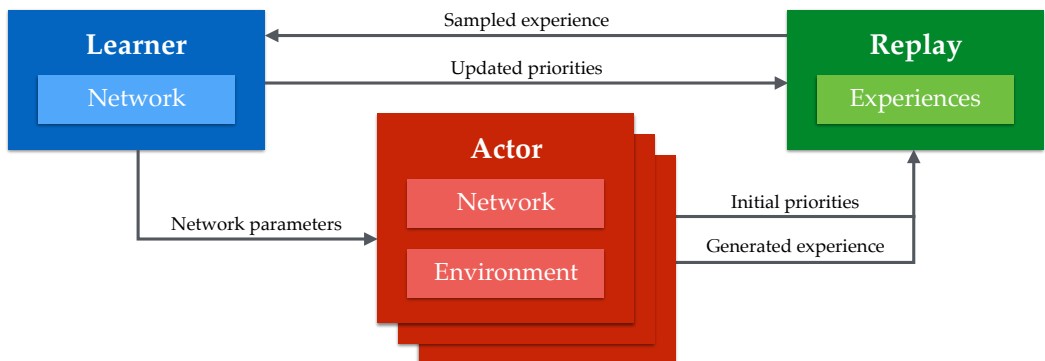

Figure 1: The Ape-X architecture in a nutshell: multiple actors, each with its own instance of the environment, generate experience, add it to a shared experience replay memory, and compute initial priorities for the data. The (single) learner samples from this memory and updates the network and the priorities of the experience in the memory. The actors' networks are periodically updated with the latest network parameters from the learner.

---

**Algorithm 1** Actor

1: **procedure** ACTOR($B, T$) ▷ Run agent in environment instance, storing experiences.
2:  $\theta_0 \leftarrow$ LEARNER.PARAMETERS( ) ▷ Remote call to obtain latest network parameters.
3:  $s_0 \leftarrow$ ENVIRONMENT.INITIALIZE( ) ▷ Get initial state from environment.
4:  **for** $t = 1$ **to** $T$ **do**
5:    $a_{t-1} \leftarrow \pi_{\theta_{t-1}}(s_{t-1})$ ▷ Select an action using the current policy.
6:    $(r_t, \gamma_t, s_t) \leftarrow$ ENVIRONMENT.STEP($a_{t-1}$) ▷ Apply the action in the environment.
7:    LOCALBUFFER.ADD(($s_{t-1}, a_{t-1}, r_t, \gamma_t$)) ▷ Add data to local buffer.
8:    **if** LOCALBUFFER.SIZE( ) $\geq B$ **then** ▷ In a background thread, periodically send data to replay.
9:      $\tau \leftarrow$ LOCALBUFFER.GET($B$) ▷ Get buffered data (e.g. batch of multi-step transitions).
10:     $p \leftarrow$ COMPUTEPRIORITIES($\tau$) ▷ Calculate priorities for experience (e.g. absolute TD error).
11:     REPLAY.ADD($\tau, p$) ▷ Remote call to add experience to replay memory.
12:   **end if**
13:   PERIODICALLY($\theta_t \leftarrow$ LEARNER.PARAMETERS()) ▷ Obtain latest network parameters.
14:  **end for**
15: **end procedure**

---

**Algorithm 2** Learner

1: **procedure** LEARNER($T$) ▷ Update network using batches sampled from memory.
2:  $\theta_0 \leftarrow$ INITIALIZENETWORK( )
3:  **for** $t = 1$ **to** $T$ **do** ▷ Update the parameters $T$ times.
4:    $id, \tau \leftarrow$ REPLAY.SAMPLE( ) ▷ Sample a prioritized batch of transitions (in a background thread).
5:    $l_t \leftarrow$ COMPUTELOSS($\tau; \theta_t$) ▷ Apply learning rule; e.g. double Q-learning or DDPG
6:    $\theta_{t+1} \leftarrow$ UPDATEPARAMETERS($l_t; \theta_t$)
7:    $p \leftarrow$ COMPUTEPRIORITIES( ) ▷ Calculate priorities for experience, (e.g. absolute TD error).
8:    REPLAY.SETPRIORITY($id, p$) ▷ Remote call to update priorities.
9:    PERIODICALLY(REPLAY.REMOVETOFIT()) ▷ Remove old experience from replay memory.
10:  **end for**
11: **end procedure**

---

As in Gorila (Nair et al., 2015), we decompose the standard deep reinforcement learning algorithm into two parts, which run concurrently with no high-level synchronization. The first part consists of stepping through an environment, evaluating a policy implemented as a deep neural network, and storing the observed data in a replay memory. We refer to this as *acting*. The second part consists of sampling batches of data from the memory to update the policy parameters. We term this *learning*.

In principle, both acting and learning may be distributed across multiple workers. In our experiments, hundreds of actors run on CPUs to generate data, and a single learner running on a GPU samples the most useful experiences (Figure 1). Pseudocode for the actors and learners is shown in Algorithms 1 and 2. Updated network parameters are periodically communicated to the actors from the learner.

In contrast to Nair et al. (2015), we use a shared, centralized replay memory, and instead of sampling uniformly, we prioritize, to sample the most useful data more often. Since priorities are shared, high priority data discovered by any actor can benefit the whole system. Priorities can be defined in various ways, depending on the learning algorithm; two instances are described in the next sections.

In Prioritized DQN (Schaul et al., 2016) priorities for new transitions were initialized to the maximum priority seen so far, and only updated once they were sampled. This does not scale well: due to the large number of actors in our architecture, waiting for the learner to update priorities would result in a myopic focus on the most recent data, which has maximum priority by construction. Instead, we take advantage of the computation the actors in Ape-X are already doing to evaluate their local copies of the policy, by making them also compute suitable priorities for new transitions online. This ensures that data entering the replay has more accurate priorities, at no extra cost.

Sharing experiences has certain advantages compared to sharing gradients. Low latency communication is not as important as in distributed SGD, because experience data becomes outdated less rapidly than gradients, provided the learning algorithm is robust to off-policy data. Across the system, we take advantage of this by batching all communications with the centralized replay, increasing the efficiency and throughput at the cost of some latency. With this approach it is even possible for actors and learners to run in different data-centers without limiting performance.

Finally, by learning off-policy (cf. Sutton & Barto, 1998; 2017), we can further take advantage of Ape-X's ability to combine data from many distributed actors, by giving the different actors different exploration policies, broadening the diversity of the experience they jointly encounter. As we will see in the results, this can be sufficient to make progress on difficult exploration problems.

## 3.1 APE-X DQN

The general framework we have described may be combined with different learning algorithms. First, we combined it with a variant of DQN (Mnih et al., 2015) with some of the components of Rainbow (Hessel et al., 2017). More specifically, we used double Q-learning (van Hasselt, 2010; van Hasselt et al., 2016) with multi-step bootstrap targets (cf. Sutton, 1988; Sutton & Barto, 1998; 2017; Mnih et al., 2016) as the learning algorithm, and a dueling network architecture (Wang et al., 2016) as the function approximator $q(\cdot, \cdot, \boldsymbol{\theta})$.

This results in computing for all elements in the batch the loss $l_t(\boldsymbol{\theta}) = \frac{1}{2}(G_t - q(S_t, A_t, \boldsymbol{\theta}))^2$ with

$$G_t = \underbrace{R_{t+1} + \gamma R_{t+2} + \ldots + \gamma^{n-1} R_{t+n} + \gamma^n \overbrace{q(S_{t+n}, \underset{a}{\operatorname{argmax}} q(S_{t+n}, a, \boldsymbol{\theta}), \boldsymbol{\theta}^-)}^{\text{double-Q bootstrap value}}}_{\text{multi-step return}},$$

where $t$ is a time index for an experience sampled from the replay starting with state $S_t$ and action $A_t$, and $\boldsymbol{\theta}^-$ denotes parameters of the *target network* (Mnih et al., 2015), a slow moving copy of the online parameters. Multi-step returns are truncated if the episode ends in fewer than $n$ steps.

In principle, Q-learning variants are off-policy methods, so we are free to choose the policies we use to generate data. However, in practice, the choice of behaviour policy does affect both exploration and the quality of function approximation. Furthermore, we are using a multi-step return with no off-policy correction, which in theory could adversely affect the value estimation. Nonetheless, in Ape-X DQN, each actor executes a different policy, and this allows experience to be generated from a variety of strategies, relying on the prioritization mechanism to pick out the most effective experiences. In our experiments, the actors use $\epsilon$-greedy policies with different values of $\epsilon$. Low $\epsilon$ policies allow exploring deeper in the environment, while high $\epsilon$ policies prevent over-specialization.

## 3.2 APE-X DPG

To test the generality of the framework we also combined it with a continuous-action policy gradient system based on DDPG (Lillicrap et al., 2016), an implementation of deterministic policy gradients Silver et al. (2014) also similar to older methods (Werbos, 1990; Prokhorov & Wunsch, 1997), and tested it on continuous control tasks from the DeepMind Control Suite (Tassa et al., 2018).

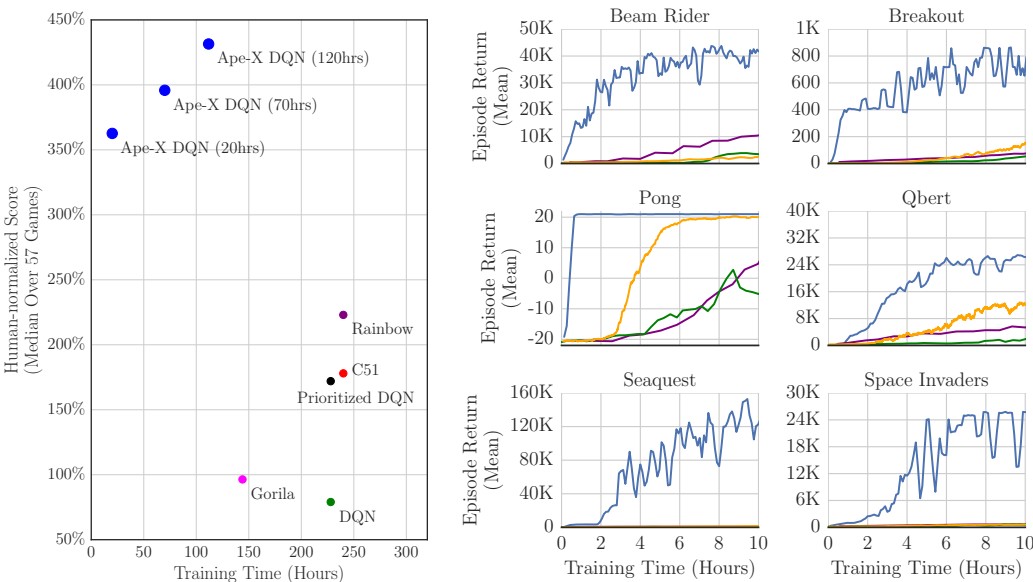

Figure 2: Left: Atari results aggregated across 57 games, evaluated from random no-op starts. Right: Atari training curves for selected games, against baselines. Blue: Ape-X DQN with 360 actors; Orange: A3C; Purple: Rainbow; Green: DQN. See appendix for longer runs over all games.

The Ape-X DPG setup is similar to Ape-X DQN, but the actor's policy is now represented explicitly by a separate policy network, in addition to the Q-network. The two networks are optimized separately, by minimizing different losses on the sampled experience. We denote the policy and Q-network parameters by $\phi$ and $\psi$ respectively, and adopt the same convention as above to denote target networks. The Q-network outputs an action-value estimate $q(s, a, \psi)$ for a given state $s$, and multi-dimensional action $a \in \mathbb{R}^m$. It is updated using temporal-difference learning with a multi-step bootstrap target. The Q-network loss can be written as $l_t(\psi) = \frac{1}{2}(G_t - q(S_t, A_t, \psi))^2$, where

$$G_t = \underbrace{R_{t+1} + \gamma R_{t+2} + \ldots + \gamma^{n-1}R_{t+n} + \gamma^n q(S_{t+n}, \pi(S_{t+n}, \phi^-), \psi^-)}_{\text{multi-step return}} .$$

The policy network outputs an action $A_t = \pi(S_t, \phi) \in \mathbb{R}^m$. The policy parameters are updated using policy gradient *ascent* on the estimated Q-value, using gradient $\nabla_\phi q(S_t, \pi(S_t, \phi), \psi)$ — note that this depends on the policy parameters $\phi$ only through the action $A_t = \pi(S_t, \phi)$ that is input to the critic network. Further details of the Ape-X DPG algorithm are available in the appendix.

# 4 EXPERIMENTS

## 4.1 ATARI

In our first set of experiments we evaluate Ape-X DQN on Atari, and show state of the art results on this standard reinforcement learning benchmark. We use 360 actor machines (each using one CPU core) to feed data into the replay memory as fast as they can generate it; approximately 139 frames per second (FPS) each, for a total of ~50K FPS, which corresponds to ~12.5K transitions (because of a fixed action repeat of 4). The actors batch experience data locally before sending it to the replay: up to 100 transitions may be buffered at a time, which are then sent asynchronously in batches of $B = 50$. The learner asynchronously prefetches up to 16 batches of 512 transitions, and computes updates for 19 such batches each second, meaning that gradients are computed for ~9.7K transitions per second on average. To reduce memory and bandwidth requirements, observation data is compressed using a PNG codec when sent and when stored in the replay. The learner decompresses data as it prefetches it, in parallel with computing and applying gradients. The learner also asynchronously handles any requests for parameters from actors.

| Algorithm | Training Time | Environment Frames | Resources (per game) | Median (no-op starts) | Median (human starts) |
|---|---|---|---|---|---|
| Ape-X DQN | 5 days | 22800M | 376 cores, 1 GPU [a] | **434%** | **358%** |
| Rainbow | 10 days | 200M | 1 GPU | 223% | 153% |
| Distributional (C51) | 10 days | 200M | 1 GPU | 178% | 125% |
| A3C | 4 days | — | 16 cores | — | 117% |
| Prioritized Dueling | 9.5 days | 200M | 1 GPU | 172% | 115% |
| DQN | 9.5 days | 200M | 1 GPU | 79% | 68% |
| Gorila DQN [c] | ~4 days | — | unknown [b] | 96% | 78% |
| UNREAL [d] | — | 250M | 16 cores | 331% [d] | 250% [d] |

Table 1: Median normalized scores across 57 Atari games. [a] Tesla P100. [b] >100 CPUs, with a mixed number of cores per CPU machine. [c] Only evaluated on 49 games. [d] Hyper-parameters were tuned per game.

Actors copy the network parameters from the learner every 400 frames (~2.8 seconds). Each actor $i \in \{0, ..., N-1\}$ executes an $\epsilon_i$-greedy policy where $\epsilon_i = \epsilon^{1+\frac{i}{N-1}\alpha}$ with $\epsilon = 0.4$, $\alpha = 7$. Each $\epsilon_i$ is held constant throughout training. The episode length is limited to 50000 frames during training.

The capacity of the shared experience replay memory is soft-limited to 2 million transitions: adding new data is always permitted, to not slow down the actors, but every 100 learning steps any excess data above this capacity threshold is removed en masse, in FIFO order. The median actual size of the memory is 2035050. Data is sampled according to proportional prioritization, with a priority exponent of 0.6 and an importance sampling exponent set to 0.4.

In Figure 2, on the left, we compare the median human normalized score across all 57 games to several baselines: DQN, Prioritized DQN, Distributional DQN (Bellemare et al., 2017), Rainbow, and Gorila. In all cases the performance is measured at the end of training under the *no-op starts* testing regime (Mnih et al., 2015). On the right, we show initial learning curves (taken from the greediest actor) for a selection of 6 games (full learning curves for all games are in the appendix). Given that Ape-X can harness substantially more computation than most baselines, one might expect it to train faster. Figure 2 shows that this was indeed the case. Perhaps more surprisingly, our agent achieved a substantially higher final performance.

In Table 1 we compare the median human-normalized performance of Ape-X DQN on the Atari benchmark to corresponding metrics as reported for other baseline agents in their respective publications. Whenever available we report results both for *no-op starts* and for *human starts*. The human-starts regime (Nair et al., 2015) corresponds to a more challenging generalization test, as the agent is initialized from random starts drawn from games played by human experts. Ape-X's performance is higher than the performance of any of the baselines according to both metrics.

## 4.2 CONTINUOUS CONTROL

In a second set of experiments we evaluated Ape-X DPG on four continuous control tasks. In the *manipulator* domain the agent must learn to bring a ball to a specified location. In the *humanoid* domain the agent must learn to control a humanoid body to solve three distinct tasks of increasing complexity: Standing, Walking and Running. Since here we learn from features, rather than from pixels, the observation space is much smaller than it is in the Atari domain. We therefore use small, fully-connected networks (details in the appendix). With 64 actors on this domain, we obtain ~14K total FPS (the same number of transitions per second; here we do not use action repeats). We process 86 batches of 256 transitions per second, or ~22K transitions processed per second.

Figure 3 shows that Ape-X DPG achieved very good performance on all four tasks. The figure shows the performance of Ape-X DPG for different numbers of actors: as the number of actors increases our agent becomes increasingly effective at solving these problems rapidly and reliably, outperforming a standard DDPG baseline trained for over 10 times longer. A parallel paper (Barth-Maron et al., 2018) builds on this work by combining Ape-X DPG with distributional value functions, and the resulting algorithm is successfully applied to further continuous control tasks.

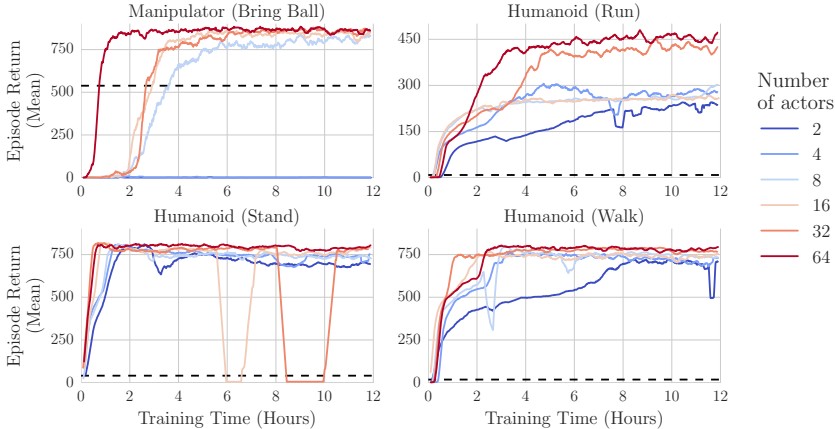

Figure 3: Performance of Ape-X DPG on four continuous control tasks, as a function of wall clock time. Performance improves as we increase the numbers of actors. The black dashed line indicates the maximum performance reached by a standard DDPG baseline over 5 days of training.

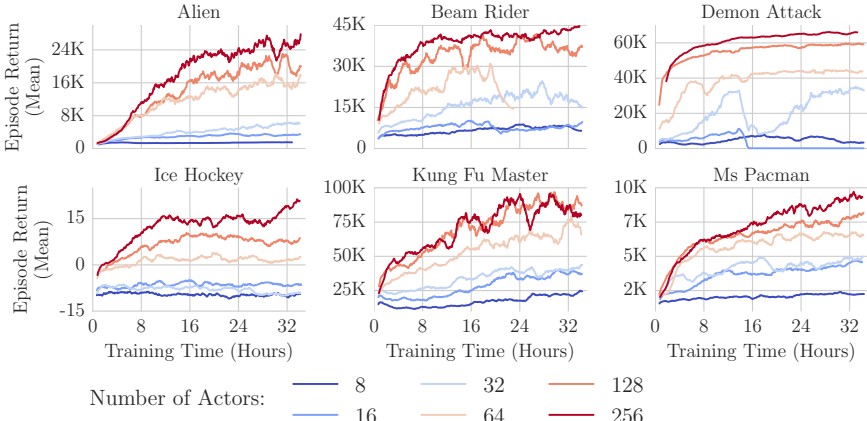

Figure 4: Scaling the number of actors. Performance consistently improves as we scale the number of actors from 8 to 256, note that the number of learning updates performed does not depend on the number of actors.

## 5 ANALYSIS

In this section we describe additional Ape-X DQN experiments on Atari that helped improve our understanding of the framework, and we investigate the contribution of different components.

First, we investigated how the performance scales with the number of actors. We trained our agent with different numbers of actors (8, 16, 32, 64, 128 and 256) for 35 hours on a subset of 6 Atari games. In all experiments we kept the size of the shared experience replay memory fixed at 1 million transitions. Figure 4 shows that the performance consistently improved as the number of actors increased. The appendix contains learning curves for additional games, and a comparison of the scalability of the algorithm with and without prioritized replay. It is perhaps surprising that performance improved so substantially purely by increasing the number of actors, without changing the rate at which the network parameters are updated, the structure of the network, or the update rule. We hypothesize that the proposed architecture helps with a common deep reinforcement learning failure mode, in which the policy discovered is a local optimum in the parameter space, but not a global one, e.g., due to insufficient exploration. Using a large number of actors with varying amounts of exploration helps to discover promising new courses of action, and prioritized replay ensures that when this happens, the learning algorithm focuses its efforts on this important information.

Next, we investigated varying the capacity of the replay memory (see Figure 5). We used a setup with 256 actors, for a median of ~37K total environment frames per second (approximately ~9K transitions). With such a large number of actors, the contents of the memory is replaced much faster than in most DQN-like agents. We observed a small benefit to using a larger replay capacity. We

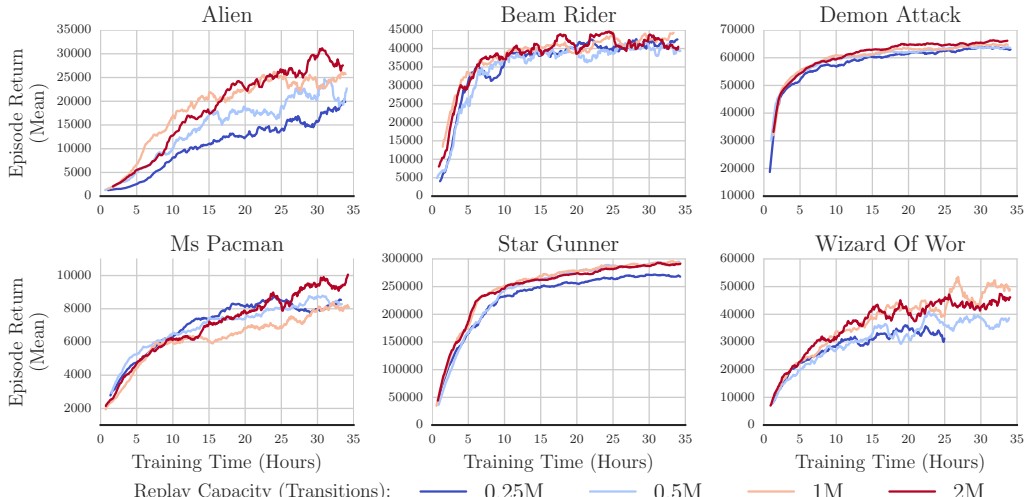

Figure 5: Varying the capacity of the replay. Agents with larger replay memories perform better on most games. Each curve corresponds to a single run, smoothed over 20 points. The curve for Wizard Of Wor with replay size 250K is incomplete because training diverged; we did not observe this with the other replay sizes.

hypothesize this is due to the value of keeping some high priority experiences around for longer and replaying them. As above, a single learner machine trained the network with median 19 batches per second, each of 512 transitions, for a median of ∼9.7K transitions processed per second.

Finally, we ran additional experiments to disentangle potential effects of two confounding factors in our scalability analysis: recency of the experience data in the replay memory, and diversity of the data-generating policies. The full description of these experiments is confined to the appendix; to summarize, neither factor alone is sufficient to explain the performance we see. We therefore conclude that the results are due substantially to the positive effects of gathering more experience data; namely better exploration of the environment and better avoidance of overfitting.

## 6 CONCLUSION

We have designed, implemented, and analyzed a distributed framework for prioritized replay in deep reinforcement learning. This architecture achieved state of the art results in a wide range of discrete and continuous tasks, both in terms of wall-clock learning speed and final performance.

In this paper we focused on applying the Ape-X framework to DQN and DPG, but it could also be combined with any other off-policy reinforcement learning update. For methods that use temporally extended sequences (e.g., Mnih et al., 2016; Wang et al., 2017), the Ape-X framework may be adapted to prioritize sequences of past experiences instead of individual transitions.

Ape-X is designed for regimes in which it is possible to generate large quantities of data in parallel. This includes simulated environments but also a variety of real-world applications, such as robotic arm farms, self-driving cars, online recommender systems, or other multi-user systems in which data is generated by many instances of the same environment (c.f. Silver et al., 2013). In applications where data is costly to obtain, our approach will not be directly applicable. With powerful function approximators, overfitting is an issue: generating more training data is the simplest way of addressing it, but may also provide guidance towards data-efficient solutions.

Many deep reinforcement learning algorithms are fundamentally limited by their ability to explore effectively in large domains. Ape-X uses a naive yet effective mechanism to address this issue: generating a diverse set of experiences and then identifying and learning from the most useful events. The success of this approach suggests that simple and direct approaches to exploration may be feasible, even for synchronous agents.

Our architecture illustrates that distributed systems are now practical both for research and, potentially, large-scale applications of deep reinforcement learning. We hope that the algorithms, architecture, and analysis we have presented will help to accelerate future efforts in this direction.

ACKNOWLEDGMENTS

We would like to acknowledge the contributions of our colleagues at DeepMind, whose input and support has been vital to the success of this work. Thanks in particular to Tom Schaul, Joseph Modayil, Sriram Srinivasan, Georg Ostrovski, Josh Abramson, Todd Hester, Jean-Baptiste Lespiau, Alban Rrustemi and Dan Belov.

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

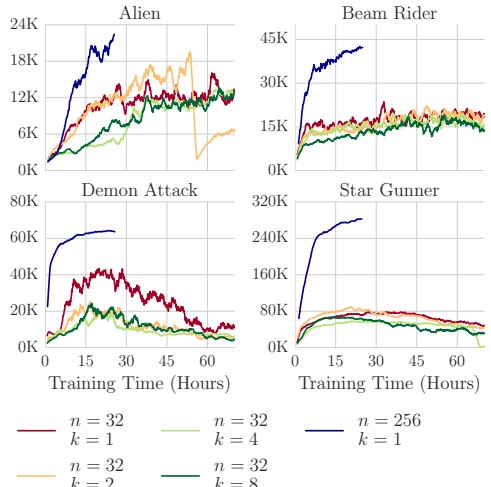

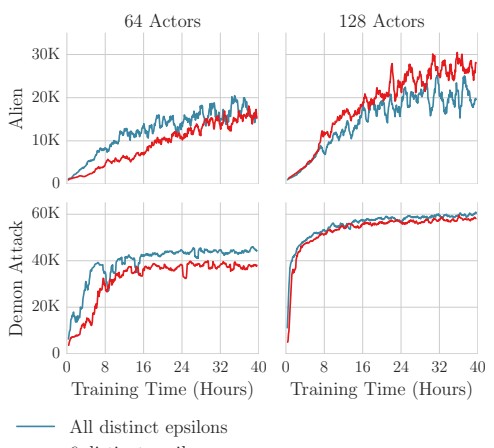

Figure 6: Testing whether improved performance is caused by recency alone: $n$ denotes the number of actors, $k$ the number of times each transition is replicated in the replay. The data in the run with $n = 32$, $k = 8$ is therefore as recent as the data in the run with $n = 256$, $k = 1$, but performance is not as good.

Figure 7: Varying the data-generating policies: Red: fixed set of 6 values for $\epsilon$. Blue: full range of values for $\epsilon$. In both cases, the curve plotted is from a separate actor that does not add data to the replay memory, and which follows an $\epsilon$-greedy policy with $\epsilon = 0.00164$.

## A    RECENCY OF EXPERIENCE

In our main experiments we do not change the size of the replay memory in proportion to the number of actors, so by changing the number of actors we also increased the rate at which the contents of the replay memory is replaced. This means that in the experiments with more actors, transitions in the replay memory are more recent: they are generated by following policies whose parameters are closer to version of the parameters being optimized by the learner, and in this sense they are more on-policy. Could this alone be sufficient to explain the improved performance? If so, we might be able to recover the results without needing a large number of actor machines. To test this, we constructed an experiment wherein we replicate the rate at which the contents of the replay memory is replaced in the 256-actor experiments, but instead of actually using 256 actors, we use 32 actors but add each transition they generate to the replay memory 8 times over. In this setup, the contents of the replay memory is similarly generated by policies with a recent version of the network parameters: the only difference is that the data is not as diverse as in the 256-actor case. We observe (see Figure 6) that this does not recover the same performance, and therefore conclude that the recency of the experience alone is not sufficient to explain the performance of our method. Indeed, we see that adding the same data multiple times can sometimes harm performance, since although it increases recency this comes at the expense of diversity.

Note: in principle, duplicating the added data in this fashion has a similar effect to reducing the capacity of the replay memory, and indeed, our results with a smaller replay memory in Figure 5 do corroborate the finding. However, we test also by duplicating the data primarily in order to exclude any effects arising from the implementation. In particular, in contrast to simply reducing the replay capacity, duplicating each data point means that the computational demands on the replay server in these runs are the same as when we use the corresponding number of real actors.

## B    VARYING THE DATA-GENERATING POLICIES

Another factor that could conceivably contribute to the scalability of our algorithm is the fact that each actor has a different $\epsilon$. To determine the extent to which this impacts upon the performance, we ran an experiment (see Figure 7) with some simple variations on the mechanism we use to choose the policies that generate the data we train on. The first alternative we tested is to choose a small fixed set of 6 values for $\epsilon$, instead of the full range that we typically use. In this test, we use prioritized

replay as normal, and we find that the results with the full range of $\epsilon$ are overall slightly better. However, it is not essential for achieving good results within our distributed framework.

## C ATARI: ADDITIONAL DETAILS

The frames received from the environment are preprocessed on the actor side with the standard transformations introduced by DQN. This includes greyscaling, frame stacking, repeating actions 4 times, and clipping rewards to $[-1, 1]$.

The learner waits for at least 50000 transitions to be accumulated in the replay before starting learning. We use a Centered RMSProp optimizer with a learning rate of 0.00025 / 4, decay of 0.95, epsilon of 1.5e-7, and no momentum to minimize the multi-step loss (with $n = 3$). Gradient norms are clipped to 40. The target network used in the loss calculation is copied from the online network every 2500 training batches. We use the same network as in the Dueling DDQN agent.

## D CONTINUOUS CONTROL: ADDITIONAL DETAILS

The critic network has a layer with 400 units, followed by a tanh activation, followed by another layer of 300 units. The actor network has a layer with 300 units, followed by a tanh activation, followed by another layer of 200 units. The gradient used to update the actor network is clipped to $[-1, 1]$, element-wise. Training uses the Adam optimizer (Kingma & Ba (2014)) with learning rate of 0.0001. The target network used in the loss calculation is copied from the online network every 100 training batches.

Replay sampling priorities are set according to the absolute TD error as given by the critic, and are sampled by the learner using proportional prioritized sampling (see appendix F) with priority exponent $\alpha_{\text{sample}} = 0.6$. To maintain a fixed replay capacity of $10^6$, transitions are periodically evicted using proportional prioritized sampling, with priority exponent $\alpha_{\text{evict}} = -0.4$. This is a different strategy for removing data than in the Atari experiments, which simply removed the oldest data first - it remains to be seen which is superior.

Unlike the original DPG algorithm which applies autocorrelated noise sampled from a Ornstein-Uhlenbeck process (Uhlenbeck & Ornstein (1930)), we apply exploration noise to each action sampled from a normal distribution with $\sigma = 0.3$. Evaluation is performed using the noiseless deterministic policy. Hyperparameters are otherwise as per DQN.

Benchmarking was performed in two continuous control domains ((a) Humanoid and (b) Manipulator, see Figure 8) implemented in the MuJoCo physics simulator (Todorov et al. (2012)). Humanoid is a humanoid walker with action, state and observation dimensionalities $|\mathcal{A}| = 21$, $|\mathcal{S}| = 55$ and $|\mathcal{O}| = 67$ respectively. Three Humanoid tasks were considered: walk (reward for exceeding a minimum velocity), run (reward proportional to movement speed) and stand (reward proportional to standing height). Manipulator is a 2-dimensional planar arm with $|\mathcal{A}| = 2$, $|\mathcal{S}| = 22$ and $|\mathcal{O}| = 37$, which receives reward for catching a randomly-initialized moving ball.

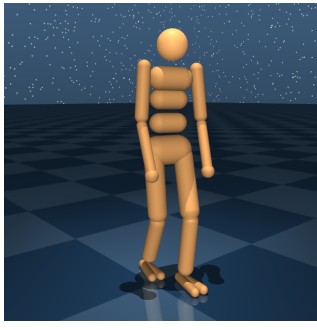
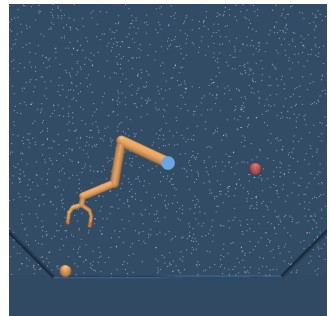

(a) Humanoid domain.           (b) Manipulator domain.

Figure 8: Continuous control domains considered for benchmarking Ape-X DPG: (a) Humanoid, and (b) Manipulator. All tasks simulated in the MuJoCo physics simulator (Todorov et al. (2012)).

# E    TUNING

On Atari, we performed some limited tuning of the learning rate and batch size: we found that larger batch sizes contribute significantly to performance, when using many actors. We tried batch sizes from $\{32, 128, 256, 512, 1024\}$, seeing clear benefits up to 512. We attempted increasing the learning rate to 0.00025 with the larger batch sizes but this destabilized training on some games. We also tried a lower learning rate of 0.00025 / 8, but this did not reliably improve results.

Likewise for continuous control, we experimented with batch sizes $\{32, 128, 256, 512, 1024\}$ and learning rates from $10^{-3}$ to $10^{-5}$. We also experimented with the prioritization exponents $\alpha$ from 0.0 to 1.0, with results proving essentially consistent within the range $[0.3, 0.7]$ (beyond 0.7, training would sometimes become unstable and diverge).

For the experiments with many actors, we set the period for updating network parameters on the actors to be high enough that the learner was not overloaded with requests, and we set the number of transitions that are locally accumulated on each actor to be high enough that the replay server would not be overloaded with network traffic, but we did not otherwise tune those parameters and have not observed them to have significant impact on the learning dynamics.

# F    IMPLEMENTATION

The following section makes explicit some of the more practical details that may be of interest to anyone wishing to implement a similar system.

**Data Storage**    The algorithm is implemented using TensorFlow (Abadi et al., 2016). Replay data is kept in a distributed in-memory key-value store implemented using custom TensorFlow ops, similar to the lookup ops available in core TensorFlow. The ops allow adding, reading, and removing batches of Tensor data efficiently.

**Sampling Data**    We also implemented ops for efficiently maintaining and sampling from a prioritized distribution over the keys, using the algorithm for proportional prioritization described in Schaul et al. (2016). The probability of sampling a transition is $p_k^\alpha / \sum_k p_k^\alpha$ where $p_k$ is the priority of the transition with key $k$. The exponent $\alpha$ controls the amount of prioritization, and when $\alpha = 0$ uniform sampling is recovered. The proportional variant sets priority $p_k = |\delta_k|$ where $\delta_k$ is the TD error for transition $k$. Whenever a batch of data is added to or removed from the store, or is processed by the learner, this distribution is correspondingly updated, recording any change to the set of valid keys and the priorities associated with them.

A background thread on the learner fetches batches of sampled data from the remote replay and decompresses it using the learner's CPU, in parallel with the gradients being computed on the GPU. The fetched data is buffered in a TensorFlow queue, so that the GPU always has data available to train on.

**Adding Data**    In order to efficiently construct $n$-step transition data, each actor maintains a circular buffer of capacity $n$ containing tuples $(S_t, A_t, R_{t:t+B}, \gamma_{t:t+B}, q(S_t, *))$, where $B$ is the current size of the buffer. With each step, the new data is appended and the accumulated per-step discounts $\gamma_{t:t+B}$ and partial returns $R_{t:t+B}$ for all entries in the buffer are updated. If the buffer has reached its capacity, $n$, then its first element may be combined with the latest state $S_{t+n}$ and value estimates $q(S_{t+n})$ to produce a valid $n$-step transition (with accompanying Q-values).

However, instead of being directly added to the remote replay memory on each step, the constructed transitions $(S_t, A_t, R_{t:t+B}, \gamma_{t:t+B}, S_{t+n}, q(S_t, *), q(S_{t+n}, *))$ are first stored in a local TensorFlow queue, in order to reduce the number of requests to the replay server. The queue is periodically flushed, at which stage the absolute $n$-step TD-errors (and thus the initial priorities) for the queued transitions are computed in batch, using the buffered Q-values to avoid recomputation. The Q-value estimates from which the initial priorities are derived are therefore based on the actor's copy of the network parameters at the time the corresponding state was obtained from the environment, rather than the latest version on the learner. These Q-values need not be stored after this, since the learner does not require them, although they can be helpful for debugging.

A unique key is assigned to each transition, which records which actor and environment step it came from, and the dequeued transition tuples are stored in the remote replay memory. As mentioned in the previous section, the remote sampling distribution is immediately updated with the newly added keys and the corresponding initial priorities computed by the actor. Note that, since we store both the start and the end state with each transition, we are storing some data twice: this costs more RAM, but simplifies the code.

**Contention**    It is important that the replay server be able to handle all requests in a timely fashion, in order to avoid slowing down the whole system. Possible bottlenecks include CPU, network bandwidth, and any locks protecting the shared data. In our experiments we found CPU to be the main bottleneck, but this was resolved by ensuring all requests and responses use sufficiently large batches. Nonetheless, it is advisable to consider all of these potential performance concerns when designing such systems.

**Asynchronicity**    In our framework, since acting and learning proceed with no synchronization, and performance depends on both, it can be misleading to consider performance with reference to only one of these. For example, the results after a given total number of environment frames have been experienced are highly dependent on the number of updates the learner has performed in that time. For this reason it is important to monitor and report the speeds of all parts of the system and to consider them when analyzing results.

**Failure Tolerance**    In distributed systems with many workers, it is inevitable that interruptions or failures will occur, either due to occasional hardware issues or because shared resources are needed by higher priority jobs. All stateful parts of the system therefore must periodically save their work and be able to resume where they left off when restarted. In our system, actors may be interrupted at any time and this will not prevent continued learning, albeit with a temporarily reduced rate of new data entering the replay memory. If the replay server is interrupted, the data it contains is discarded, and upon resuming, the memory is refilled quickly by the actors. In this event, to avoid overfitting, the learner will pause training briefly, until the minimum amount of data has once again been accumulated. If the learner is interrupted, progress will stall until it resumes.

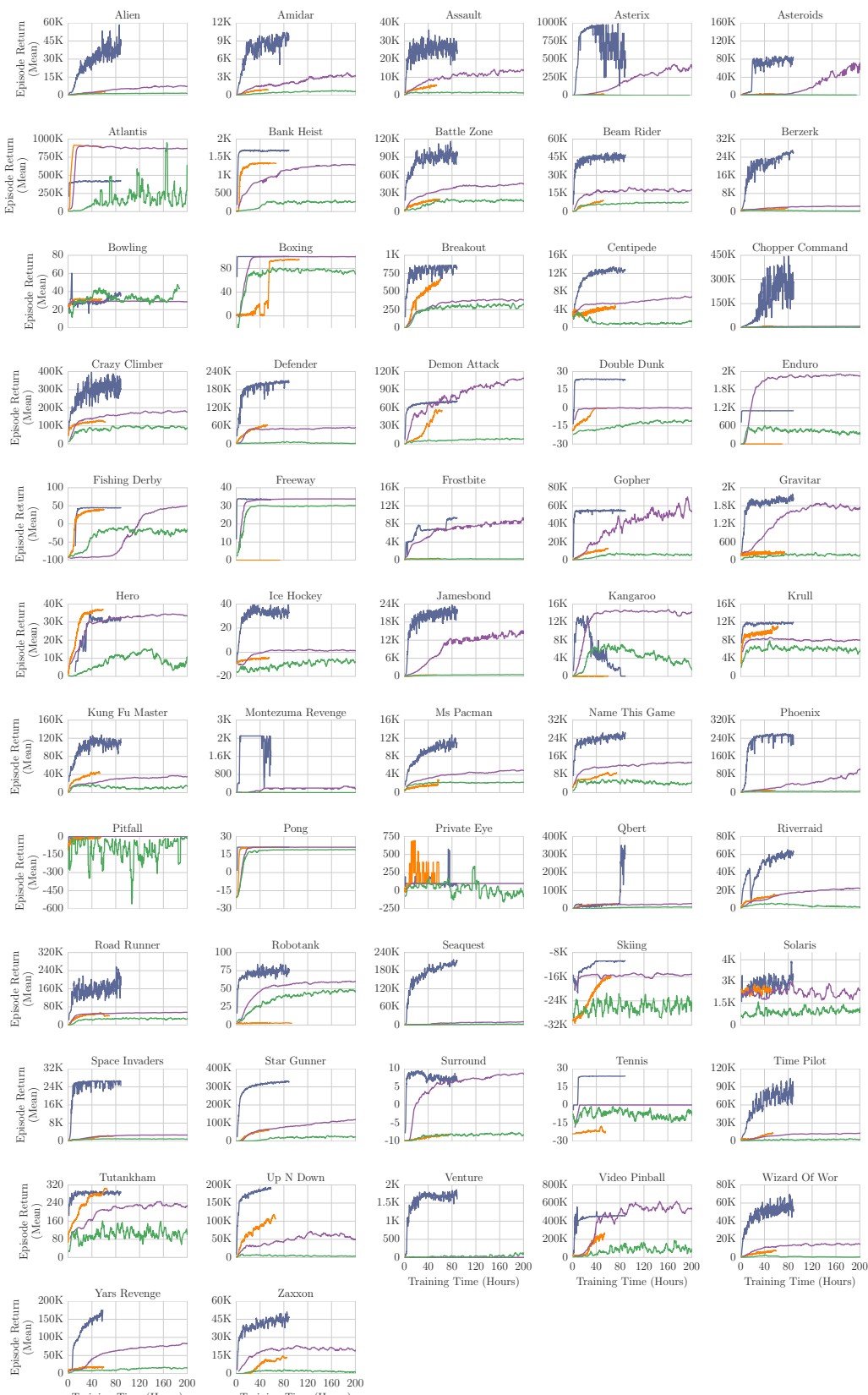

Figure 9: Training curves for 57 Atari games (performance against wall clock time). Green: DQN baseline. Purple: Rainbow baseline. Orange: A3C baseline. Blue: Ape-X DQN with 360 actors, 1 replay server and 1 Tesla P100 GPU learner. The anomaly in Riverraid is due to an infrastructure error.

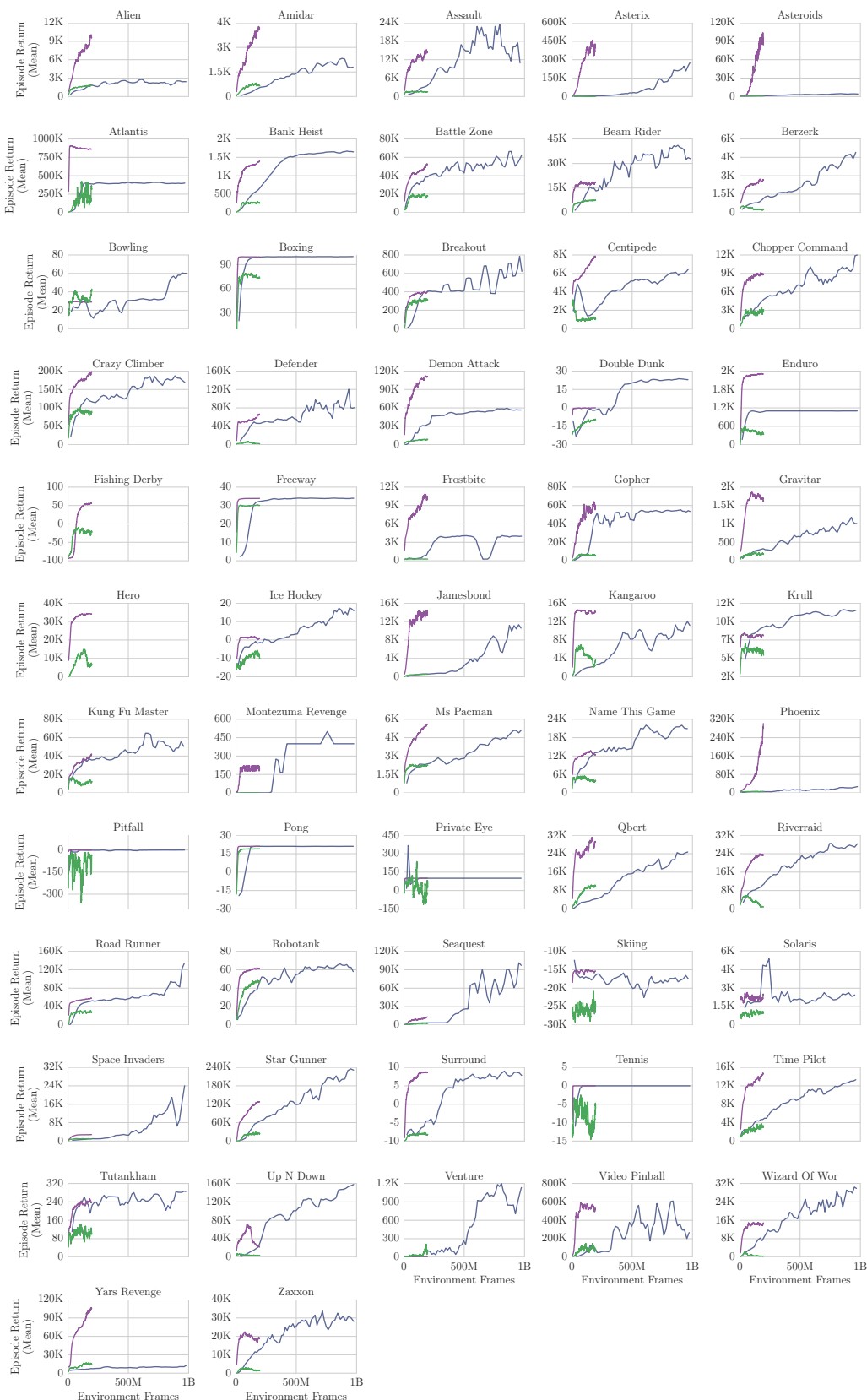

Figure 10: Training curves for 57 Atari games (performance against environment frames). Only the first billion frames are shown, corresponding to 5-6 hours of training for Ape-X. Green: DQN baseline. Purple: Rainbow baseline. Blue: ApeX-DQN with 360 actors, 1 replay server and 1 Tesla P100 GPU learner.

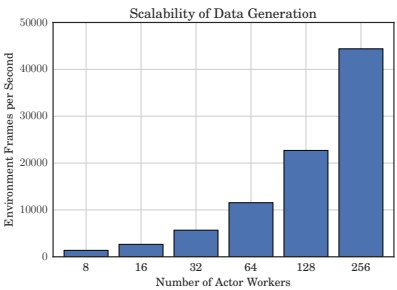

Figure 11: Speed of data generation scales linearly with the number of actors.

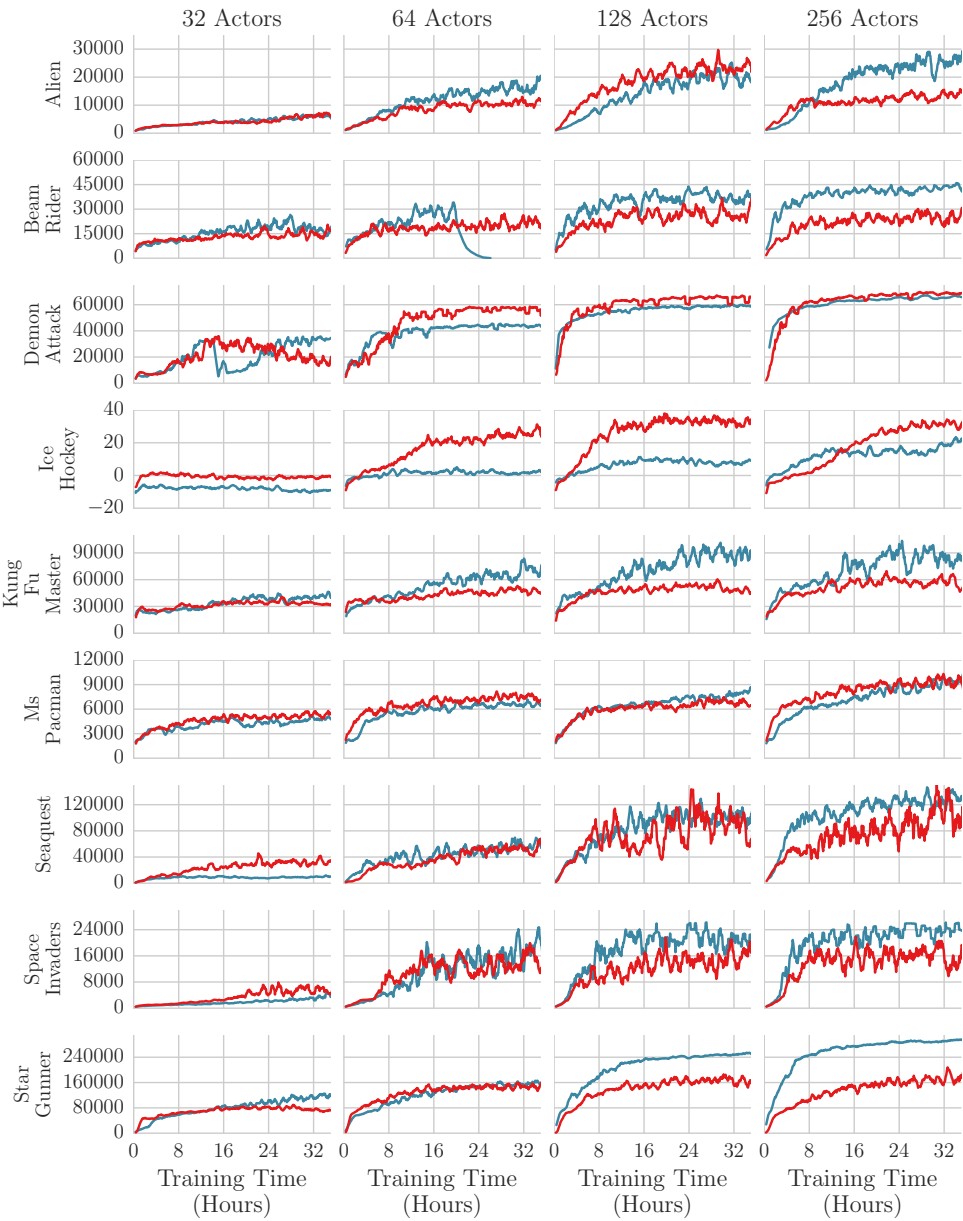

Figure 12: Training curves showing performance against wall clock time for various numbers of actors on a selection of Atari games. Blue: prioritized replay, with learning rate 0.00025 / 4. Red: uniform replay, with learning rate 0.00025. For both prioritized and uniform, we tried both of these learning rates and selected the best. Both variants benefit from larger numbers of actors, but prioritized can better take advantage of the increased amount of data. In the 256-actor run, prioritized is equal or better in 7 of 9 games.

| Game | No-op starts | Human starts |
|---|---|---|
| alien | 40,804.9 | 17,731.5 |
| amidar | 8,659.2 | 1,047.3 |
| assault | 24,559.4 | 24,404.6 |
| asterix | 313,305.0 | 283,179.5 |
| asteroids | 155,495.1 | 117,303.4 |
| atlantis | 944,497.5 | 918,714.5 |
| bank_heist | 1,716.4 | 1,200.8 |
| battle_zone | 98,895.0 | 92,275.0 |
| beam_rider | 63,305.2 | 72,233.7 |
| berzerk | 57,196.7 | 55,598.9 |
| bowling | 17.6 | 30.2 |
| boxing | 100.0 | 80.9 |
| breakout | 800.9 | 756.5 |
| centipede | 12,974.0 | 5,711.6 |
| chopper_command | 721,851.0 | 576,601.5 |
| crazy_climber | 320,426.0 | 263,953.5 |
| defender | 411,943.5 | 399,865.3 |
| demon_attack | 133,086.4 | 133,002.1 |
| double_dunk | 23.5 | 22.3 |
| enduro | 2,177.4 | 2,042.4 |
| fishing_derby | 44.4 | 22.4 |
| freeway | 33.7 | 29.0 |
| frostbite | 9,328.6 | 6,511.5 |
| gopher | 120,500.9 | 121,168.2 |
| gravitar | 1,598.5 | 662.0 |
| hero | 31,655.9 | 26,345.3 |
| ice_hockey | 33.0 | 24.0 |
| jamesbond | 21,322.5 | 18,992.3 |
| kangaroo | 1,416.0 | 577.5 |
| krull | 11,741.4 | 8,592.0 |
| kung_fu_master | 97,829.5 | 72,068.0 |
| montezuma_revenge | 2,500.0 | 1,079.0 |
| ms_pacman | 11,255.2 | 6,135.4 |
| name_this_game | 25,783.3 | 23,829.9 |
| phoenix | 224,491.1 | 188,788.5 |
| pitfall | -0.6 | -273.3 |
| pong | 20.9 | 18.7 |
| private_eye | 49.8 | 864.7 |
| qbert | 302,391.3 | 380,152.1 |
| riverraid | 63,864.4 | 49,982.8 |
| road_runner | 222,234.5 | 127,111.5 |
| robotank | 73.8 | 68.5 |
| seaquest | 392,952.3 | 377,179.8 |
| skiing | -10,789.9 | -11,359.3 |
| solaris | 2,892.9 | 3,115.9 |
| space_invaders | 54,681.0 | 50,699.3 |
| star_gunner | 434,342.5 | 432,958.0 |
| surround | 7.1 | 5.5 |
| tennis | 23.9 | 23.0 |
| time_pilot | 87,085.0 | 71,543.0 |
| tutankham | 272.6 | 127.7 |
| up_n_down | 401,884.3 | 347,912.2 |
| venture | 1,813.0 | 935.5 |
| video_pinball | 565,163.2 | 873,988.5 |
| wizard_of_wor | 46,204.0 | 46,897.0 |
| yars_revenge | 148,594.8 | 131,701.1 |
| zaxxon | 42,285.5 | 37,672.0 |

Table 2: Scores obtained by Ape-X DQN in final evaluation, under the standard no-op starts and human starts regimes. In some games the scores are higher than in the training curves: this is because the maximum episode length is shorter during training. 