# OpenReview forum: "Distributed Prioritized Experience Replay"
_ICLR.cc/2018/Conference — Accept (Poster)_

### Official Review · AnonReviewer2 · 2017-11-10
**A new way of parallelizing distributed deep RL**

**Rating:** 9
**Confidence:** 4

**Review:**

This paper examines a distributed Deep RL system in which experiences, rather than gradients, are shared between the parallel workers and the centralized learner. The experiences are accumulated into a central replay memory and prioritized replay is used to update the policy based on the diverse experience accumulated by all the of the workers. Using this system, the authors are able to harness much more compute to learn very high quality policies in little time. The results very convincingly show that Ape-X far outperforms competing algorithms such as recently published Rainbow.

It’s hard to take issue with a paper that has such overwhelmingly convincing experimental results. However, there are a couple additional experiments that would be quite nice:
•	In order to understand the best way for training a distributed RL agent, it would be nice to see a side-by-side comparison of systems for distributed gradient sharing (e.g. Gorila) versus experience sharing (e.g. Ape-X).
•	It would be interesting to get a sense of how Ape-X performs as a function of the number of frames it has seen, rather than just wall-clock time. For example, in Table 1, is Ape-X at 200M frames doing better than Rainbow at 200M frames?

Pros:
•	Well written and clear.
•	Very impressive results.
•	It’s remarkable that Ape-X preforms as well as it does given the simplicity of the algorithm.

Cons:
•	Hard to replicate experiments without the deep computational pockets of DeepMind.

---

> ### Author Response · Authors · 2018-01-01
> **Thank you very much for the review.**
>
> Thank you very much for the review. This is a good summary of the paper.
>
> Q1: on side-by-side comparison of systems for distributed gradient sharing (e.g. Gorila) versus experience sharing (e.g. Ape-X).
>
> A thorough exploration and comparison of these approaches would be valuable, but we believe that fairly and rigorously investigating this large space of possible designs is likely to be a complex topic unto itself, and it would not be possible to do it justice in this paper. Performance comparisons of such systems will likely depend on practical factors such as network latency (due to stale gradients or straggling workers) as well as model size and the size of the observation data (since this will affect the throughput across the distributed system). Ultimately we believe that distributed gradient sharing and distributed experience sharing will prove complementary, but that the nuances of how to optimally combine them will therefore depend on not only the domain but also the nature and distribution of the available computational resources.
>
> Q2: on how Ape-X performs as a function of the number of frames it has seen, rather than just wall-clock time.
>
> In case you missed it, Figure 10 in the Appendix includes plots of performance against number of frames seen for the first billion frames, with comparisons against Rainbow and DQN. Note, however, that in all of these algorithms, the amount of experience replay per environment step may be varied, and this factor can have a significant effect on such results.

---

### Official Review · AnonReviewer3 · 2017-11-27
**A clear proof that parallelizing DQN computations works**

**Rating:** 7
**Confidence:** 4

**Review:**

A parallel aproach to DQN training is proposed, based on the idea of having multiple actors collecting data in parallel, while a single learner trains the model from experiences sampled from a central replay memory. Experiments on Atari game playing and two MuJoCo continuous control tasks show significant improvements in terms of training time and final performance compared to previous baselines.

The core idea is pretty straightforward but the paper does a very good job at demonstrating that it works very well, when implemented efficiently over a large cluster (which is not trivial). I also appreciate the various experiments to analyze the impact of several settings (instead of just reporting a new SOTA). Overall I believe this is definitely a solid contribution that will benefit both practitioners and researchers... as long as they got the computational resources to do so!

There are essentially two more things I would have really liked to see in this paper (maybe for future work?):
- Using all Rainbow components
- Using multiple learners (with actors cycling between them for instance)
Sharing your custom Tensorflow implementation of prioritized experience replay would also be a great bonus!

Minor points:
- Figure 1 does not seem to be referenced in the text
- « In principle, Q-learning variants are off-policy methods » => not with multi-step unless you do some kind of correction! I think it is important to mention it even if it works well in practice (just saying « furthermore we are using a multi-step return » is too vague)
- When comparing the Gt targets for DQN vs DPG it strikes me that DPG uses the delayed weights phi- to select the action, while DQN uses current weights theta. I am curious to know if there is a good motivation for this and what impact this can have on the training dynamics.
- In caption of Fig. 5 25K should be 250K
- In appendix A why duplicate memory data instead of just using a smaller memory size?
- In appendix D it looks like experiences removed from memory are chosen by sampling instead of just removing the older ones as in DQN. Why use a different scheme?
- Why store rewards and gamma’s at each time step in memory instead of just the total discounted reward?
- It would have been better to re-use the same colors as in Fig. 2 for plots in the appendix
- Would Fig. 10 be more interesting with the full plot and a log scale on the x axis?

---

> ### Author Response · Authors · 2018-01-01
> **Thank you for the thorough review!**
>
> Thank you for the thorough review! This is a good summary of the paper.
>
> Q1: on using all Rainbow components and on using multiple learners.
>
> These are both interesting directions which we agree may help to boost performance even further. For this paper, we felt that adding extra components would distract from the finding that it is possible to improve results significantly by scaling up, even with a relatively simple algorithm.
>
> Q2: on sharing the custom Tensorflow implementation of prioritized experience replay.
>
> We would love to share an implementation, as we have found prioritization to be a consistently helpful component and would like to see it more widely used, but when we looked into this we realized that the current version depends on a library that would require a significant amount of engineering work to open source - so unfortunately we can’t commit to it at this time. However, we will bear this in mind for future versions.
>
> Q3: on multi-step Q-learning not being off-policy.
>
> We’ll try to clarify this.
>
> Q4: on which weights to use for action selection in DQN vs DPG target computations.
>
> Interesting observation - setting aside the multi-step modification, our Ape-X DQN targets follow the approach described in [1] directly, whilst the Ape-X DPG targets are the same as those described in [2]. For the sake of simplicity in this paper, we were motivated not to deviate from the previously described update rules, in order to focus primarily on the improvements that could be obtained by our modifications to the method of generating and selecting the training data.
>
> However, to answer your question on a more technical level, in [2], the motivation given for the use of the target network weights to select the action when computing target values is that “the target values are constrained to change slowly, greatly improving the stability of learning” - the authors of [2] further note that they “found that having both a target µ’ and Q’ was required to have stable targets y_i in order to consistently train the critic without divergence. This may slow learning, since the target network delays the propagation of value estimations. However, in practice we found this was greatly outweighed by the stability of learning”.
>
> In [1], the update is modified in order to reduce overestimation resulting from the maximization step in Q-learning; they note that “the selection of the action, in the argmax, is still due to the online weights θ_t. This means that, as in Q-learning, we are still estimating the value of the greedy policy according to the current values, as defined by θ_t. However, we use the second set of weights θ’_t to fairly evaluate the value of this policy”.
>
> We have not yet re-evaluated these choices to determine whether the conclusions still hold in our new system. However, note that in DDPG (and thus also in Ape-X DDPG) there is no maximization step in the critic update, since we are using temporal-difference learning to update the critic instead of Q-learning - so the decoupling of action selection from evaluation used in Double Q Learning does not apply directly anyway.
>
> We do not claim that these combinations of learning rules and target networks are necessarily the optimal ones, but we hope that this helps to explain the rationale behind the choices used in this paper.
>
> [1] https://arxiv.org/abs/1509.06461
> [2] https://arxiv.org/abs/1509.02971
>
> Q5: in appendix A why duplicate memory data instead of just using a smaller memory size?
>
> Conceptually, it would be indeed be sufficient to use a smaller memory to investigate this effect; in fact our results in Figure 5 begin to do this - but we wanted to corroborate the finding by also measuring it in a different way. For implementation reasons, the two approaches are not guaranteed to be equivalent: for example, duplicating the data that each actor adds increases the computational load on the replay server, whereas using a smaller memory size does not. During development we noticed that in very extreme cases, many actors adding large volumes of data to the replay memory could overwhelm it, causing a slowdown in sampling which would affect the performance of the learner and thus the overall results.
>
> In our experiments in Appendix A where we sought to determine whether recency of data was the reason for our observed scalability results, we wanted to make certain that the load on the replay server in the duplicated-data experiments would be the same as in the experiments with the corresponding numbers of real actors, to ensure a fair comparison. In practice, we did not find that we were running into any such contention issues in these experiments, and the results from Figure 5 do agree with those in Appendix A. However, we felt that it was still helpful to include both of the results in order to cover this aspect thoroughly. We will add a note explaining this.

---

> > ### Author Response · Authors · 2018-01-01
> > **(continued)**
> >
> > Q6: In appendix D it looks like experiences removed from memory are chosen by sampling instead of just removing the older ones as in DQN. Why use a different scheme?
> >
> > We believe that this prioritized removal scheme may improve upon the usual FIFO removal approach, since it allows high priority data to remain in memory for longer. We have not yet re-run the Atari experiments with this newer modification, due to the significant resource requirements - we apologize for the discrepancy and we will add some explanation to make this more explicit.
> >
> > Q7: Why store rewards and gamma’s at each time step in memory instead of just the total discounted reward?
> >
> > To clarify, we are storing the sum of discounted rewards accumulated across each multi-step transition, and the product of gammas across each multi-step transition. While this is not the only way to do it, these are cheap to compute online on each actor worker, and are sufficient to be able to compute up-to-date target values easily on the learner. We will make this more explicit in the implementation section in the appendix.
> >
> > Q8: Would Fig. 10 be more interesting with the full plot and a log scale on the x axis?
> >
> > We tried this but decided it was too difficult to read that way, unfortunately... Since the data is from the same experiment as Figure 9 and the rate of data generation is approximately constant, the information that would be available in a full plot can largely be inferred from Figure 9, though.
> >
> > Q9: on the other minor points (Fig 1 reference, Fig 5 caption, and Fig 2 plot colors)
> >
> > Thanks! We’ll fix these oversights.

---

> > > ### Comment · AnonReviewer3 · 2018-01-10
> > > **Re: Thank you for the thorough review!**
> > >
> > > Thank you for the detailed response and paper revision

---

### Official Review · AnonReviewer1 · 2017-11-30
**A somewhat trivial extension of Prioritized Experience Replay by adding parallelization in actor algorithm**

**Rating:** 6
**Confidence:** 3

**Review:**

This paper proposes a distributed architecture for deep reinforcement learning at scale, specifically, focusing on adding parallelization in actor algorithm in Prioritized Experience Replay framework. It has a very nice introduction and literature review of Prioritized experience replay and also suggested to parallelize the actor algorithm by simply adding more actors to execute in parallel, so that the experience replay can obtain more data for the learner to sample and learn. Not surprisingly, as this framework is able to learn from way more data (e.g. in Atari), it outperforms the baselines, and Figure 4 clearly shows the more actors we have the better performance we will have.

While the strength of this paper is clearly the good writing as well as rigorous experimentation, the main concern I have with this paper is novelty. It is in my opinion a somewhat trivial extension of the previous work of Prioritized experience replay in literature; hence the challenge of the work is not quite clear. Hence, I feel adding some practical learnings of setting up such infrastructure might add more flavor to this paper, for example.

---

> ### Author Response · Authors · 2018-01-01
> **Thank you for your comments and for this helpful suggestion.**
>
> Thank you for your comments and for this helpful suggestion. Our work is indeed closely related to the previous work on Prioritized Experience Replay. In achieving our reported results in practice, there was considerable challenge in two aspects: firstly, in the engineering work necessary to run the algorithm at a large scale, and secondly, in the discovery through empirical experimentation of a) the necessary algorithmic extensions to the prior work upon which we built, and b) the best way in which to combine them in practice. The point is well taken that the difficulty of this may not have been evident from our description, since in the paper we opted to focus more on our final architecture and results, believing this to be of greater interest to most readers. Indeed, we covered the implementation only briefly in the Appendix, and, as you note, we did not discuss our practical learnings in much depth. We are happy to hear that this is also of interest and we will gladly expand upon this section to provide further information and advice.

---

### Author Response · Authors · 2018-01-05
**Updated revision**

The updated revision includes the changes mentioned in our responses to the reviews. Thanks again to all reviewers for their valuable comments. In addition to the minor fixes and clarifications suggested, we have expanded the implementation section with some of our practical findings, which we hope adds further value to this work for anybody interested in building similar systems.

There is only one additional change in the new revision that was not previously discussed: a small fix to our description of the per-actor epsilons used in our Atari experiments.

---

### Decision · Program_Chairs · 2018-01-29
**ICLR 2018 Conference Acceptance Decision**

**Decision:**

Accept (Poster)

**Comment:**

meta score: 8

The paper present a distributed architecture using prioritized experience replay for deep reinforcement learning.  It is well-written and the experimentation is extremely strong.  The main issue is the originality - technically, it extends previous work in a limited way;  the main contribution is practical, and this is validated by the experiments.  The experimental support is such that the paper has meaningful conclusions and will surely be of interest to people working in the field.  Thus I would say it is comfortably over the acceptance threshold.

Pros:
 - good motivation and literature review
 - strong experimentation
 - well-written and clearly presented
 - details in the appendix are very helpful
Cons:
 - possibly limited originality in terms of modelling advances